# Involvement of the Retinal Pigment Epithelium in the Development of Retinal Lattice Degeneration

**DOI:** 10.3390/ijms21197347

**Published:** 2020-10-05

**Authors:** Hiroshi Mizuno, Masanori Fukumoto, Takaki Sato, Taeko Horie, Teruyo Kida, Hidehiro Oku, Kimitoshi Nakamura, Denan Jin, Shinji Takai, Tsunehiko Ikeda

**Affiliations:** 1Department of Ophthalmology, Osaka Medical College, Takatsuki-City, Osaka 569-8686, Japan; synchroboys1007@gmail.com (H.M.); opt094@osaka-med.ac.jp (M.F.); opt147@osaka-med.ac.jp (T.S.); opt168@osaka-med.ac.jp (T.H.); opt038@osaka-med.ac.jp (T.K.); opt025@osaka-med.ac.jp (H.O.); 2Nakamura Eye Clinic, Matsumoto-City, Nagano 390-0811, Japan; nakamura-ganka01@tuba.ocn.ne.jp; 3Department of Innovative Medicine, Graduate School of Medicine, Osaka Medical College, Takatsuki-City, Osaka 569-8686, Japan; pha012@osaka-med.ac.jp (D.J.); pha010@osaka-med.ac.jp (S.T.)

**Keywords:** retinal lattice degeneration, retinal detachment, retinal pigment epithelium (RPE), glial fibrillary acidic protein (GFAP), retinal pigment epithelium-specific protein 65 kDa (RPE65), cytokeratin (CK)

## Abstract

Lattice degeneration involves thinning of the retina that occurs over time. Here we performed an immunohistological study of tissue sections of human peripheral retinal lattice degeneration to investigate if retinal pigment epithelium (RPE) cells are involved in the pathogenesis of this condition. In two cases of retinal detachment with a large tear that underwent vitreous surgery, retinal lattice degeneration tissue specimens were collected during surgery. In the obtained specimens, both whole mounts and horizontal section slices were prepared, and immunostaining was then performed with hematoxylin and antibodies against glial fibrillary acidic protein (GFAP), RPE-specific protein 65 kDa (RPE65), pan-cytokeratin (pan-CK), and CK18. Hematoxylin staining showed no nuclei in the center of the degenerative lesion, thus suggesting the possibility of the occurrence of apoptosis. In the degenerative lesion specimens, GFAP staining was observed in the center, RPE65 staining was observed in the slightly peripheral region, and pan-CK staining was observed in all areas. However, no obvious CK18 staining was observed. In a monkey retina used as the control specimen of a normal healthy retina, no RPE65 or pan-CK staining was observed in the neural retina. Our findings suggest that migration, proliferation, and differentiation of RPE cells might be involved in the repair of retinal lattice degeneration.

## 1. Introduction

Peripheral retinal degeneration is characterized by various types of lesions, such as those associated with lattice degeneration, cobblestone degeneration, and cystoid degeneration [1,2,3]. Retinal lattice degeneration is a type of vitreoretinal degeneration associated with atrophic lesions of the retina and overlaying vitreous liquefaction observed in the equatorial region of the ocular fundus [4,5]. Retinal lattice degeneration is thought to be involved in the development of rhegmatogenous retinal detachment in approximately 30% of the cases, making this degeneration clinically very important [6]. Numerous reports have indicated that, histopathologically, the degenerated area of retinal lattice degeneration is thinned and devoid of photoreceptors, and is replaced by glial cells in all retinal layers [4,5]. Moreover, the degenerative lesion is often pigmented, and the thickness of the retinal pigment epithelium (RPE) corresponding to the degenerated area is simultaneously uneven and may be partially lost [4,7]. Although previous studies have reported the migration of macrophages with melanin pigment, no changes in the choroid are usually observed [7]. These findings suggest that the migration and proliferation of RPE cells may be involved in the pathogenesis of retinal lattice degeneration. However, to the best of our knowledge, there have been no previous studies investigating whether the markers of RPE cells are expressed in the lesions of retinal lattice degeneration.

Retinal pigment epithelium-specific protein 65 kDa (RPE65) is a specific marker of RPE cells, and is reportedly involved in retinoid metabolism, one of the functions of retinal pigment epithelial cells [8]. Cytokeratin (CK) is one of the intermediate filaments that form the cytoskeleton of epithelial cells, and it is specifically expressed in the retinal pigment epithelium [9].

The purpose of this present study was to perform RPE65 and CK immunostaining on retinal tissues containing specimens of retinal lattice degenerative lesions collected during vitreous surgery, and to report our novel findings on the pathogenesis of retinal lattice degeneration.

## 2. Results

### 2.1. Collection of the Retinal Lattice Degeneration Specimens

In both cases, retinal lattice degeneration specimens at the approximate width and length of 1 mm and 3 mm, respectively, were collected (Figure 1a,b), and the papilla diameter was approximately 1.5 mm in the two cases. No intraoperative complications were observed during the collection of the lesions. After resection of the peripheral retinal lattice degeneration, we performed pneumatic retinopexy, endophotocoagulation, and a gas tamponade with 20% sulfur hexafluoride. In both cases, the retina was attached after surgery, and the postoperative courses were favorable.

### 2.2. Hematoxylin Staining of Retinal Lattice Degeneration

Although no nuclei were observed in the center of each retinal lattice degeneration specimen, diffuse nuclei were detected in the peripheral regions (Figure 2, blue arrowheads). However, no nuclei were observed in the central cells (Figure 2, black arrowheads).

### 2.3. Glial Fibrillary Acidic Protein (GFAP), RPE65, and CK Immunostaining of the Human Retinal Lattice Degeneration Specimens

In Case 1, the tissue specimen stained positive for GFAP, and was more intensely stained in the slightly central area. In addition, pigment-containing cells were observed in some of the areas that were less intensely stained (Figure 3a). For RPE65, the entire tissue specimen stained positive, while the margins of the central area without nuclei were more intensely stained (Figure 3b). For pan-CK, the tissue was almost consistently stained positive (Figure 3c), whereas it was little stained with CK18 (Figure 3d). The negative control is shown in Figure 3e.

In Case 2, the entire tissue specimen stained positive for RPE65, as in Case 1, yet the central area of the tissue with a few nuclei was less intensely stained (Figure 4a). For pan-CK, the tissue was almost consistently stained positive (Figure 4b). The number of nuclei that stained positive for RPE65 in an area of 50 μm^2^ was approximately five to six. However, the number of nuclei that stained positive for pan-CK in that same 50 μm^2^ area was approximately 20 to 30. The negative control (i.e., without the RPE65 and pan-CK antibodies) is shown in Figure 4c.

### 2.4. RPE65 and CK Immunostaining of the Normal Healthy Retina of the Monkey Eyes

Since a normal healthy retina could not be ethically harvested from human subjects, a normal healthy retina of a cynomolgus monkey was used to perform immunostaining with RPE65 and pan-CK. The neural retina was not stained with RPE65 or pan-CK (Figure 5a,b).

## 3. Discussion

Retinal lattice degenerations clinically appear as spindle-shaped or band-like, thin retinal lesions with a well-demarcated margin, and the lesion is often circumferentially located around the equatorial region of the ocular fundus. Atypical lesions display a lattice-work-type pattern, in which blood vessels appear as white lines that traverse the degenerative lesions. A degenerative lesion measures 0.5 to 1.5 papilla diameters in width, whereas its length widely varies. Some lesions measure less than 1 papilla in diameter, and there are also lesions that measure 1 to 2 quadrants. The lesion occurs in approximately 10% of normal healthy eyes, and its familial occurrence is well-known. The periphery of a degenerative lesion slightly protrudes, whereas the center is depressed due to the retina being thinned. The vitreous body above a degenerative lesion forms a liquefaction cavity, and the membranous vitreous body adheres to the periphery of the lesion. Due to the posterior vitreous detachment, a retinal tear is likely to occur along the posterior edge of a degenerative lesion. The color of the inside of a degenerative lesion is an opaque whitish gray color, and pigments often transmigrate to various extents [4,5,6].

As described above, the degenerated area of the lesion is often pigmented, and simultaneously, the RPE corresponding to the degenerated area is of uneven thickness and may be partially lost [4,7]. When the RPE cell layer is lost, the degenerated retina comes directly into contact with the Bruch’s membrane. However, no abnormalities, such as obstruction or loss, are seen on the choroidal capillary lamina [4,5,6,7]. These findings indicate that the RPE cells are involved in the pathogenesis of retinal lattice degeneration.

RPE cells have features similar to those of primitive neural stem cells. It has been reported that when the retina is damaged, RPE cells migrate to the damage site, thus repairing the tissue [10]. Del Monte et al. [11] reported that RPE cells migrate to the retina in cases of Sanfilippo’s syndrome, in which cyst formation is observed in the retina and other tissues. Jaisley et al. [12] reported that RPE cells migrate to the degenerated neural retina, which is observed as bone spicule pigmentation in patients with retinitis pigmentosa. Similar findings are also reportedly observed in age-related macular degeneration and macular dystrophy [13,14].

In this present study, retinal lattice degenerations were intraoperatively harvested from patients with rhegmatogenous retinal detachment, and were then used to perform immunostaining with GFAP, which is a glial cell marker, and CK and RPE65, which are retinal pigment epithelial markers [15]. In such patients, with retinal detachment accompanied by a large tear at the periphery of the retinal lattice degeneration, the lesion is often intraoperatively resected with a vitreous cutter to definitively remove the vitreous traction. Thus, the procedure performed in this present study may not be detrimental to patients.

The specimen of the lattice generation stained positive for GFAP, and the central area was more intensely stained than the peripheral region (Figure 3a). GFAP is a marker of glial cells, such as reactive Müller cells and astrocytes in the adult human retina [16]. Therefore, glial cells like Müller cells or astrocytes surrounding the retinal lattice degeneration might migrate, proliferate, and replace the damaged tissues. Hematoxylin staining revealed no nuclei in the center of the retinal lattice degeneration, while scattered nuclei were observed in the periphery. The absence of nuclei in the central lesion, where GFAP was intensely stained as described above, might have indicated the occurrence of apoptosis of the proliferated glial cells. In order to verify the existence of apoptosis, a TUNEL (TdT-mediated dUTP Nick End Labeling) assay or other investigative techniques should be used. However, due to the limited amount of tissue we obtained during vitreous surgery, we were unable to perform the additional experiments in these two cases. Xu et al. reported that TUNEL-labelled characteristic DNA fragmentation of apoptosis was not observed in the photoreceptor cells in four cases of retinal lattice degeneration [17]. However, their study focused on the level of photoreceptors, so more extensive studies involving a larger number of cases will be necessary in the future.

In support of this speculation, it should be noted that in nearly all autopsy studies, the presence of newly-formed cellular components of the retinal lattice degeneration is reportedly thought to represent glial proliferation [5,18]. However, as far as we are aware, immunostaining using RPE cell markers has not been performed, and some reports indicated these new tissues might have originated from the pigment epithelium [5].

The findings in this present study revealed that the entire tissue of the retinal lattice degeneration stained positive for RPE65 and pan-CK, both markers of RPE cells [19,20], although the intensity of the staining varied (Figure 3b,c and Figure 4a,b). Both RPE65 and CKs are reportedly positive for RPE and negative for neural retina [21], except the red/green cone, which reportedly selectively expresses RPE65 [19]. We also performed immunostaining for RPE65 and pan-CK in the normal monkey retina specimen, and found that the neural retina was not stained with either marker, except for RPE65 staining in the photoreceptor layer where the red/green cone was present (Figure 5a,b). Traditionally, additional immunostaining should also be performed on other specific genes of RPE cells, such as cellular retinaldehyde-binding protein, vascular endothelial growth factor, CD68, and MERTK [22,23]. However, due to the limited amount of tissue we obtained during vitreous surgery, we were unable to perform the additional experiments in the two cases. Reactive Muller cells are reportedly negative for RPE65 and CKs [24,25,26,27]. Taken together, this evidence indicates that RPE cells might have migrated and proliferated in the retinal lattice degeneration. The pigment clumps, which were observed in the degenerative lesions, also indicated the possibility of the proliferation of migrated RPE cells. However, additional experiments, such as in situ hybridization, should be performed to verify that these markers are coming only from RPE cells.

Intraretinal migration of RPE cells reportedly occurs after the epithelial–mesenchymal transition (EMT) [28,29]. It has been shown that RPE cells transdifferentiate from an epithelial phenotype to a migratory fibroblastic phenotype during EMT [29], thus revealing that RPE reportedly lost epithelial markers, such as E-cadherin and CK18, and acquired mesenchymal markers, such as N-cadherin, α-smooth muscle actin, and vimentin [30,31,32]. Moreover, transdifferentiated RPE cells reportedly increase the expression of GFAP [27,29,33]. It should be noted that fibroblast-shaped RPE cells reportedly co-express vimentin, CKs, and GFAP in vitro [29]. Thus, the assumption that RPE cells replace the damaged tissue of the retinal lattice degeneration is consistent with our GFAP-positive immunostaining results, although it is possible that the degenerative lesion was accompanied by the proliferation of both RPE and glial cells.

In this current study, CK18, a marker of RPE cells [30], was negative in the retinal lattice degeneration (Figure 3d). It has previously been reported that among the epithelial CKs, normal RPE cells with an epithelial phenotype expressed only CK8 and CK18 [34], and that the expression of CK18 is decreased in the RPE cells after the EMT [29,30,34]. Moreover, both atrophic and hyperplastic RPE cells have reportedly been labeled with antibodies to CK7 or CK19 instead of CK18 [34]. Thus, even though the retinal lattice degeneration was negatively stained with CK18, a marker of RPE cells, it was plausible that intraretinally migrated RPE cells proliferated in the degenerative lesion.

It has long been thought that adult mammalian retinal cells lack a capacity to regenerate [35]. However, several recent studies have indicated that retinal stem cells capable of differentiating into neurons are present in the adult primate retina [36,37,38,39]. Müller and RPE cells, as well as retinal precursor cells in the foveola and peripheral retinal margin, have been considered as candidates of adult primate retinal stem cells [35,36,37,38,39,40]. In this study, we assumed that the retinal lattice degeneration might have been caused by the inhibition of the regenerative potential of retinal stem cells, possibly transdifferentiated RPE cells. In this context, two questions arise. First, why might RPE cells, instead of Müller cells, be involved in the regeneration process of retinal lattice degeneration? Second, why might the regeneration process of RPE cells fail, thus resulting in the development of a degenerative lesion? Further study is needed to elucidate the answers to these two questions.

Retinal lattice degeneration is devoid of the internal limiting membrane in the degenerative lesion, and the vitreous adjacent to the lesion is highly degenerated and liquefied. Thus, the membranous vitreous cortex firmly adheres to the margin of the degenerative lesion. These anatomical features of retinal lattice degeneration are known to be deeply involved in retinal tear formation, along with the progression of posterior vitreous detachment [41]. Many aspects as to the underlying mechanism by which this specific structure of the vitreous body appears in a degenerative lesion have yet to be elucidated. One possible hypothesis is that migrating RPE cells may produce immature collagen into the vitreous [42,43]. In general, fibroblastic cells, such as transdifferentiated RPE cells, produce immature collagen, which linearly extends without crosslinking [44]. In the vitreous base, dense collagen fibers in the vitreous extend vertically to the ciliary epithelium and retina, with the fibers inserting into the basal lamina of the ciliary body and the internal limiting membrane of the retina. Consequently, there is firm vitreoretinal adhesion in the vitreous base [45,46,47].

The collagen fibers in the vitreous body also run vertically to the retinal surface in retinal lattice degeneration, with the associated anatomical features resembling the vitreous base [4,5,6]. One possible explanation for the reason why the vertical collagen fibers appear on the retinal surface of lattice degeneration is that RPE cells migrate, proliferate, and differentiate in an attempt to repair the retinal lesion, thus producing immature vitreous fibers and collagen fibers into the vitreous overlying the retinal lattice degeneration. However, it should be noted that the above-described points still remain in the realm of speculation, and need to be investigated in future studies.

It should be noted that in the present study, only a limited number of specimens were available for testing, so additional experiments are needed in the future.

## 4. Subjects and Methods

### 4.1. Collection of the Retinal Lattice Degenerations Specimens

The protocols of this study were approved by the Ethics Committee of Osaka Medical College, Takatsuki-City, Osaka, Japan, and the study was performed in accordance with the tenets set forth in the Declaration of Helsinki. Written informed consent was obtained from all patients prior to their involvement in the study.

This study involved two patients (Case 1: a 65-year-old male; Case 2: a 58-year-old female) who underwent vitreous surgery to selectively collect retinal specimens of retinal lattice degeneration from two eyes with rhegmatogenous retinal detachment, accompanied by a large tear at the margin of retinal lattice degeneration. To collect the specimens, triamcinolone acetonide was applied around the tear in order to visualize the vitreous gel. The vitreous gel was then resected to remove, as much as possible, the vitreoretinal traction around the tear. Next, the retinal lattice degeneration tissue was held with vitreous forceps and incised at the margin of the lesion to selectively collect a tissue specimen containing the lesion. The collected retinal lattice degeneration specimens were then fixed with 4% paraformaldehyde (PFA) and embedded in VECTASHIELD (Vector Laboratories, Inc., Burlingame, CA, United States) mounting medium. Thin sections were then prepared for hematoxylin staining by slicing the specimen parallel to the retinal surface, and whole mounts were prepared for GFAP, RPE65, pan-CK, and CK18 immunostaining.

### 4.2. Hematoxylin Staining of Retinal Lattice Degeneration

In Case 1, hematoxylin staining was performed to examine the cellular components of the tissue specimens.

### 4.3. Glial Fibrillary Acidic Protein (GFAP), RPE65, Pan-CK, and CK18 Immunostaining of the Human Retinal Lattice Degeneration Specimens

GFAP staining was performed with a 1:100 dilution of the primary rabbit antibody (bs-0199 R-A647; Thermo Fisher Scientific, Inc., Fremont, CA, United States, or BioLegend, Inc., San Diego, CA, United States). RPE65 staining was performed with a 1:200 dilution of the primary anti-RPE 65 rabbit antibody (401.8 B11.3 D9, Abcam, Cambridge, United Kingdom). Pan-CK staining was performed with a 1:100 dilution of the primary anti-pan-CK rabbit antibody (C-11) (ab7753; Abcam), and CK18 staining was performed with a 1:100 dilution of the primary anti-CK18 rabbit antibody (C-04; Abcam). All stained specimens were then incubated at 4 °C for 2 days. Next, the specimens were rinsed with phosphate-buffered saline (PBS), and then incubated with biotinylated anti-rabbit immunoglobulin G (1:1000; Vector Laboratories) at room temperature (RT) for 2 h. The specimens were then rinsed with PBS and incubated with an alkaline phosphatase-labeled avidin–biotin complex (Vector Laboratories) at RT for 2 h. Next, the specimens were reacted with ImmPACT Vector-Red Substrate (Vector Laboratories). After dehydration, the specimens were encapsulated with Entellan New (Merck KGaA, Darmstadt, Germany) and then observed with a BZ-x700 All-In-One Fluorescence microscope (Keyence Corporation, Itasca, IL, United States). In Case 1, immunostaining was performed with all of the antibodies, as mentioned above. In Case 2, the antibodies against RPE65 and pan-CK were used. For comparison, we also created a negative control without these antibodies.

### 4.4. RPE65 and CK Immunostaining of the Normal Healthy Retina of Monkey Eyes

Since normal healthy human eye tissue was not available, we examined the localization of RPE65 and CK in the neural retina tissue obtained from a cynomolgus monkey eye. Briefly, a cynomolgus monkey (male, 4 years old) was euthanized, and its eyeballs were excised. The eyeballs were then fixed in 4% PFA phosphate buffer solution for 24 h. From the fixed eyeballs, the cornea and lens were resected to prepare eyecups. The vitreous body attached to the retina was then removed with tweezers. Next, the retina was embedded in VECTASHIELD, and vertical thin sections containing the macula, the optic disc, and the most peripheral retina were prepared. These sections were then immunostained with antibodies against RPE65 and pan-CK in the same manner as described above.

## Figures and Tables

**Figure 1 ijms-21-07347-f001:**
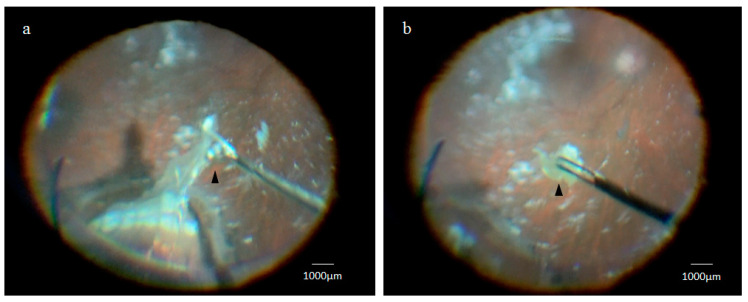
Intraoperative collection of the retinal lattice degeneration specimen. The retinal lattice degeneration specimen (black arrowhead) was held with vitreous forceps (**a**) and incised at the periphery of the lesion (black arrowhead) to selectively collect a specimen containing the lesion (**b**). This sections were then prepared by slicing the specimen parallel to the retinal surface for hematoxylin staining, and whole mounts were prepared for glial fibrillary acidic protein (GFAP), retinal pigment epithelium-specific protein 65 kDa (RPE65), pan-cytokeratin (pan-CK), and cytokeratin (CK)18 immunostaining.

**Figure 2 ijms-21-07347-f002:**
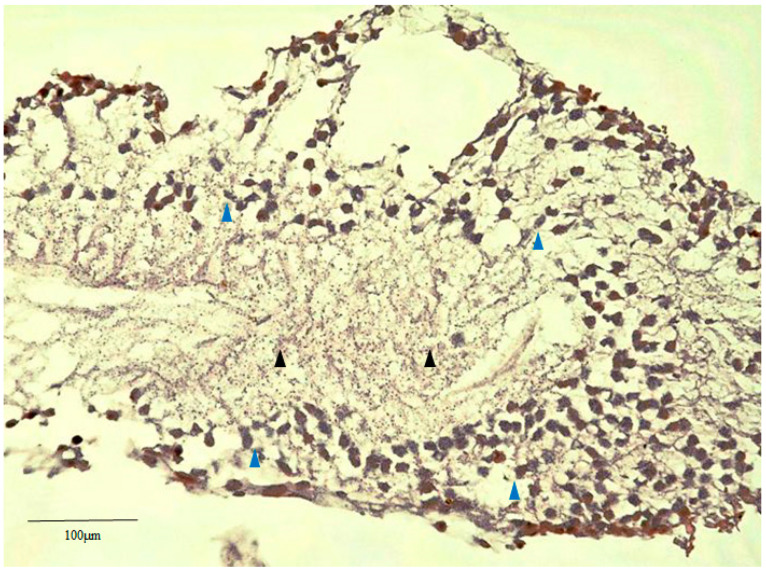
Hematoxylin staining of the obtained retinal lattice degeneration via slicing parallel to the retinal surface. No nucleus was observed in the center of the retinal lattice degeneration (black arrowheads), yet diffuse nuclei were detected in the periphery (blue arrowheads).

**Figure 3 ijms-21-07347-f003:**
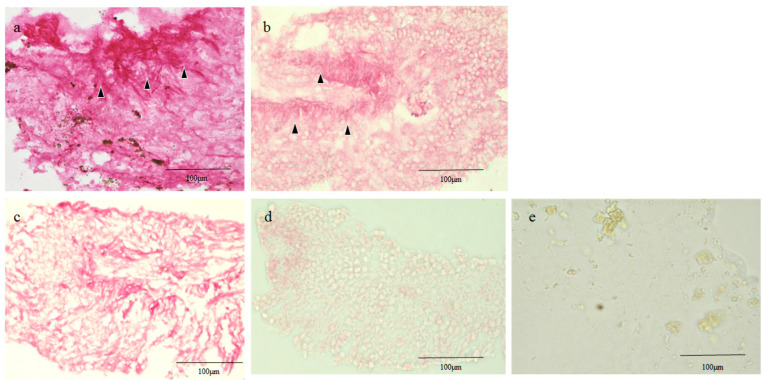
GFAP, RPE65, pan-CK, and CK18 immunostaining of the retinal lattice degeneration whole mount specimen in Case 1. The tissue was stained positive for GFAP, and the central area was slightly more intensely stained (black arrowheads) (**a**). For RPE65, the entire tissue stained positive, while the margins of the central area without nuclei were more intensely stained (black arrowheads) (**b**). For pan-CK, the tissue almost consistently stained positive (**c**), whereas it was little stained with CK18 (**d**). Negative control (**e**).

**Figure 4 ijms-21-07347-f004:**
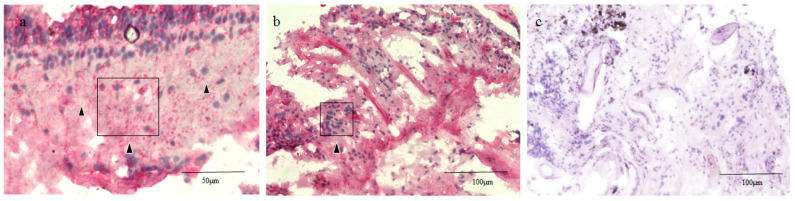
RPE65 and pan-CK immunostaining of the retinal lattice degeneration specimen in Case 2. The whole mount tissue stained positive for RPE65, whereas its central area with a few nuclei was less intensely stained (black arrowheads) (**a**). For pan-CK, the tissue almost consistently stained positive (**b**). Negative control (**c**).

**Figure 5 ijms-21-07347-f005:**
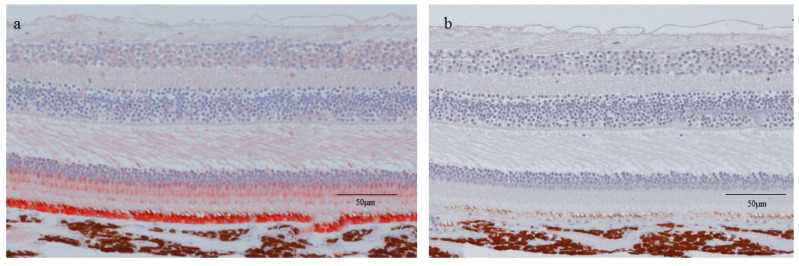
RPE65 and pan-CK immunostaining of the normal healthy retina of a monkey eye. An excised normal healthy retina of a cynomolgus monkey was used to perform the immunostaining with RPE65 (**a**) and pan-CK (**b**). The neural retina was not stained with RPE65 or pan-CK.

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
