# Peer review of "Involvement of the Retinal Pigment Epithelium in the Development of Retinal Lattice Degeneration"

_ijms, 2020, doi:10.3390/ijms21197347_

Round 1
Reviewer 1 Report
In the present study authors have identified the presence of RPE markers in retinal lattice degeneration model suggesting that migration, proliferation, and differentiation of RPE cells might be involved in the repair of retinal lattice degeneration. (1) The authors need to justify why they chose these markers (RPE65. pan-CK, GFAP), Even though a little is described in the discussion it is not enough since the whole results are supported by this immunostaining only. (2) Authors should do a TUNEL assay to show the presence of dead cells that might have led to GFAP immunostaining. (3) Authors can also do a in-situ hybridization to prove that the markers are coming only from RPE cells since the immunostaining might have got artefacts due to the nature of the tissue itself. (4) Place arrows or arrowheads in each image panel where you describe change in intensity. (5) Since in retinal lattice degeneration the structure of the whole retina is not intact and there is a loss of photoreceptors, there is also a possibility that the RPE cells do not have a structure to maintain its normal position and hence seen everywhere rather than hep in an attempt to proliferate and differentiate. It is important to show the prsence of some other RPE markers that may be involved in proliferation if authors like to support their hypothesis.Author Response
Reviewer 1’ comments
In the present study authors have identified the presence of RPE markers in retinal lattice degeneration model suggesting that migration, proliferation, and differentiation of RPE cells might be involved in the repair of retinal lattice degeneration.
(1) The authors need to justify why they chose these markers (RPE65. pan-CK, GFAP), Even though a little is described in the discussion it is not enough since the whole results are supported by this immunostaining only.
Response to Comment 1: We greatly appreciate the Reviewer’s comment. In accordance with the Reviewer’s suggestion, please note that we have now revised the Introduction section of the manuscript to include additional text as to why RPE65 and CK were used, as well as two additional references as follows:
"Retinal pigment epithelium-specific protein 65kDa (RPE65) is a specific marker of RPE cells and is reportedly involved in retinoid metabolism, one of the functions of retinal pigment epithelial cells [8]. Cytokeratin (CK) is one of the intermediate filaments that form the cytoskeleton of epithelial cells, and it is specifically expressed in the retinal pigment epithelium [9]." (Lines 60-64)
"8. Mata, N.L.; Moghrabi, W.N.; Lee, J.S.; Bui, T.V.; Radu, R.A.; Horwitz, J.; Travis, G.H. Rpe65 is a retinyl ester binding protein that presents insoluble substrate to the isomerase in retinal pigment epithelial cells. J. Biol. Chem. 2004, 279, 635-643; doi: 10.1074/jbc.M310042200.
- Labelle, P.; Reilly, C.M.; Naydan, D.K.; Labelle, A.L. Immunohistochemical characteristics of normal canine eyes. Vet. Pathol. 2012, 49, 860-869; doi: 10.1177/0300985811427152." (Lines 335-341)
(2) Authors should do a TUNEL assay to show the presence of dead cells that might have led to GFAP immunostaining.
Response to Comment 2: We greatly appreciate the Reviewer’s helpful comment. Please note that due to the limited amount of tissue that we were able to obtain during vitreous surgery, we were unable to perform the TUNEL assay in this study. As an alternative, please note that we have now added a statement citing the previous literature on apoptosis in retinal lattice degeneration, as follows:
"In order to verify the existence of apoptosis, TUNEL assay and/or other investigative techniques should be used. However, due to the limited amount of tissue we obtained during vitreous surgery, we were unable to perform the additional experiments in these two cases. Xu et al. reported that TUNEL-labelling characteristic DNA fragmentation of apoptosis was not observed in the photoreceptors cells in four cases of retinal lattice degeneration [17]. However, their study focused on the level of photoreceptors, so more extensive studies involving a larger number of cases will be necessary in the future." (Lines 157-164)
"17. Xu, G.Z.; Li, W.W.; Tso, M.O. Apoptosis in human retinal degenerations. Trans. Am. Ophthalmol. Soc. 1996, 94, 411-430." (Lines 370 and 371)
(3) Authors can also do a in-situ hybridization to prove that the markers are coming only from RPE cells since the immunostaining might have got artefacts due to the nature of the tissue itself.
Response to Comment 3: We greatly appreciate the Reviewer’s instructive comment. Please note that due to the limited amount of tissue that we were able to obtain during vitreous surgery, we were unable to perform the in-situ hybridization in this study. Please note that we have now added the following statement in the Discussion section:
"However, additional experiments, such as in-situ hybridization, should be performed to verify that these markers are coming only from RPE cells." (Lines 187-189)
(4) Place arrows or arrowheads in each image panel where you describe change in intensity.
Response to Comment 4: We greatly appreciate the Reviewer’s comment. In accordance with the Reviewer’s suggestion, please note that we have now placed arrowheads in Figures 1-4, and have revised the associated figure legends to read as follows:
"Figure 1. Intraoperative collection of the retinal lattice degeneration specimen. The retinal lattice degeneration specimen (black arrowhead) was held with vitreous forceps (a) and incised at the periphery of the lesion (black arrowhead) to selectively collect a specimen containing the lesion (b). Thin sections were then prepared by slicing the specimen parallel to the retinal surface for hematoxylin staining, and whole mounts were prepared for glial fibrillary acidic protein (GFAP), retinal pigment epithelium-specific protein 65kDa (RPE65), pan-cytokeratin (pan-CK), and cytokeratin (CK)18 immunostaining.
Figure 2. Hematoxylin staining of the obtained retinal lattice degeneration specimen via slicing parallel to the retinal surface. No nucleus was observed in the center of the retinal lattice degeneration (black arrowheads), yet diffuse nuclei were detected in the peripheral area (blue arrowheads).
Figure 3. GFAP, RPE65, pan-CK, and CK18 immunostaining of the retinal lattice degeneration whole-mount specimen in Case 1. The tissue was stained positive for GFAP, and the central area was slightly more intensely stained (black arrowheads) (a). For RPE65, the entire tissue was stained positive, while the margins of the central area without nuclei were more intensely stained (black arrowheads) (b). For pan-CK, the tissue was almost consistently stained positive (c), whereas it was little stained with CK18 (d). Negative control (e).
Figure 4. RPE65 and pan-CK immunostaining of the retinal lattice degeneration specimen in Case 2. The whole-mount tissue was stained positive for RPE65, whereas the central area with a few nuclei was less intensely stained (black arrowheads) (a). For pan-CK, the tissue was almost consistently stained positive (b). Negative control (c)." (Lines 484-509)
(5) Since in retinal lattice degeneration the structure of the whole retina is not intact and there is a loss of photoreceptors, there is also a possibility that the RPE cells do not have a structure to maintain its normal position and hence seen everywhere rather than help in an attempt to proliferate and differentiate. It is important to show the presence of some other RPE markers that may be involved in proliferation if authors like to support their hypothesis.
Response to Comment 5: We greatly appreciate the Reviewer’s instructive comment. Please note that due to the limited amount of tissue that we were able to obtain during vitreous surgery, we were unable to perform the immunostaining by some other RPE markers. As an alternative, please note that we have now revised the text and have cited previous literature on other specific genes of RPE cells as follows:
"Traditionally, additional immunostaining should also be performed on other specific genes of RPE cells, such as cellular retinaldehyde-binding protein, vascular endothelial growth factor, CD68, and MERTK [23,24]. However, due to the limited amount of tissue we obtained during vitreous surgery, we were unable to perform the additional experiments in the two cases." (Lines 179-183)
"23. Akrami, H.; Soheili, Z.S.; Sadeghizadeh, M.; Khalooghi, K.; Ahmadieh, H.; Kanavi, M.R.; Samiei, S.; Pakravesh, J. Evaluation of RPE65, CRALBP, VEGF, CD68, and tyrosinase gene expression in human retinal pigment epithelial cells cultured on amniotic membrane. Biochem. Genet. 2011, 49, 313-322; doi: 10.1007/s10528-010-9409-1.
- Haruta, M.; Sasai, Y.; Kawasaki, H.; Amemiya, K.; Ooto, S.; Kitada, M.; Suemori, H.; Nakatsuji, N.; Ide, C.; Honda, Y.; et al. In vitro and in vivo characterization of pigment epithelial cells differentiated from primate embryonic stem cells. Invest. Ophthalmol. Vis. Sci. 2004, 45, 1020-1025; doi: 10.1167/iovs.03-1034." (Lines 387-395)
Reviewer 2 Report
Ijms-935081 revision
Brief summary
In this article by Mizuno and colleagues, entitled ‘ Immunohistological study of human retinal lattice degeneration’, the authors suggest that migration, proliferation, and differentiation of RPE cells might be involved in the repair after retina degeneration. The article is well-written in an excellent English and style. I found a few typos only (see Other comments).
The origin of the retinal lattice degeneration is an important question, there is sparse information on it. The fact that the study worked with human and macaque samples further raised its value. Although the experimental design does not reach the level that is enough to draw the right conclusions from it about Muller cell and RPE migration and related repair mechanisms. There are many issues that need to be answered before this article is published and with some additional work it would be a really valuable piece of information for others in retina or ophthalmology research.
Specific comments
- My first major comment is why the authors used different stainings on the two different human specimens? Why did not they use both stainings in both of the specimens (they were big enough to resect them). They would have been comparable in this way.
- The title is too general, it gives no information and has no statements. The authors worked with RPE65 and CK, but these are not even mentioned in the title. Finding a better title is strongly suggested.
- The abstract needs to be systematically rewritten. Parts of the text should be allocated to follow the order suggested by abstract writing recommendations (MDPI: https://www.mdpi.com/authors/layout#_bookmark5) and to achieve a level of clarity.
- In the introduction, there are no details about RPE65 and CK and about the motivation of why the authors used them and which cell-types are labeled by them in the healthy retina? Please give some relevant background to help the readers to understand these.
- In the results, the first part (2.1) mostly belongs to the materials and methods section.
- In Results 2.2 the central cells did not have nuclei. Were those dead cells?
- At L70. the authors state that the central cells are more intensely stained. It is not clear from the beginning that these stained specimens are slices or whole-mounted specimens. If the latter is true it is possible that the central area is thicker, thus having more dense staining.
- There is no quantification of the staining densities in the paper. What is visible for the naked eye will also serve as a good base for quantification. Please quantify these (densities, cell numbers).
- What is the negative control of Fig3?
- What is visible in Fig5? There are no details about it in the text nor the figure legend. How it is related to lattice degeneration? The authors miss to correctly link any relation between the two.
- In the discussion, the authors assume that GFAP+ cells are activated Müller cells but they look more like astrocytes. Could they be part of a detached ILM? There is no real basis to draw such conclusions from these results on Müller cells, although RPE migration provides a feasible explanation for the RPE65/CK stainings, that I could agree with but it needs to be further investigated.
- Please define papilla diameter, since it is not widely used by non-clinicians.
- The authors should add product codes for the antibodies consequently. I could not find the codes for GFAP antibodies. Were these the same from two different companies? Were these rabbit antibodies? Have you tested the GFAP antibodies on retinal whole mounts as well to show astrocyte staining?
- The procedure of slicing and embedding of the samples is not evidently clear. For example the mounting with Vectashield and slicing afterward. Which samples were whole mounts and which were slices? Please indicate it on the figures or figure legends.
- Please recheck the methods section to maintain possible reproductibility.
Other comments
L15. The specimens
L113. RPE cells are involved…
L130. GFAP is a marker of
L139. immunostaining
L148. These pieces of evidence
L149. which were
L178. of the internal
Author Response
Reviewer 2’ comments
Brief summary
In this article by Mizuno and colleagues, entitled ‘ Immunohistological study of human retinal lattice degeneration’, the authors suggest that migration, proliferation, and differentiation of RPE cells might be involved in the repair after retina degeneration. The article is well-written in an excellent English and style. I found a few typos only (see Other comments).
The origin of the retinal lattice degeneration is an important question, there is sparse information on it. The fact that the study worked with human and macaque samples further raised its value. Although the experimental design does not reach the level that is enough to draw the right conclusions from it about Muller cell and RPE migration and related repair mechanisms. There are many issues that need to be answered before this article is published and with some additional work it would be a really valuable piece of information for others in retina or ophthalmology research.
Specific comments
(1)My first major comment is why the authors used different stainings on the two different human specimens? Why did not they use both stainings in both of the specimens (they were big enough to resect them). They would have been comparable in this way.
Response to Comment 1: We greatly appreciate the Reviewer’s cogent comment. In accordance with the Reviewer’s suggestion, please note that we have now revised the Introduction section of the manuscript to include additional text as to why RPE65 and CK were used, as well as two additional references as follows:
"Retinal pigment epithelium-specific protein 65kDa (RPE65) is a specific marker of RPE cells and is reportedly involved in retinoid metabolism, one of the functions of retinal pigment epithelial cells [8]. Cytokeratin (CK) is one of the intermediate filaments that form the cytoskeleton of epithelial cells, and it is specifically expressed in the retinal pigment epithelium [9]." (Lines 60-64)
"8. Mata, N.L.; Moghrabi, W.N.; Lee, J.S.; Bui, T.V.; Radu, R.A.; Horwitz, J.; Travis, G.H. Rpe65 is a retinyl ester binding protein that presents insoluble substrate to the isomerase in retinal pigment epithelial cells. J. Biol. Chem. 2004, 279, 635-643; doi: 10.1074/jbc.M310042200.
- Labelle, P.; Reilly, C.M.; Naydan, D.K.; Labelle, A.L. Immunohistochemical characteristics of normal canine eyes. Vet. Pathol. 2012, 49, 860-869; doi: 10.1177/0300985811427152." (Lines 335-341)
(2)The title is too general, it gives no information and has no statements. The authors worked with RPE65 and CK, but these are not even mentioned in the title. Finding a better title is strongly suggested.
Response to Comment 2: We greatly appreciate the Reviewer’s helpful comment. In accordance with the Reviewer’s suggestion, please note that we have now revised the title to read as follows:
"Involvement of the retinal pigment epithelium in the development of retinal lattice degeneration." (Lines 1-2)
(3)The abstract needs to be systematically rewritten. Parts of the text should be allocated to follow the order suggested by abstract writing recommendations (MDPI: https://www.mdpi.com/authors/layout#_bookmark5) and to achieve a level of clarity.
Response to Comment 3: We greatly appreciate the Reviewer’s instructive comment. In accordance with the Reviewer’s suggestion, please note that we have now revised the abstract as follows:
"Abstract
In this study, we performed an immunohistological investigation of tissue sections of human peripheral retinal lattice degeneration, and our findings suggest that retinal pigment epithelium (RPE) cells might be involved in the pathogenesis of this degeneration. In two cases that underwent vitreous surgery for retinal detachment with a large tear, specimens of retinal lattice degeneration were collected during surgery. In the obtained specimens, both whole mounts and slices of horizontal sections were prepared, and immunostaining was then performed with hematoxylin and antibodies against glial fibrillary acidic protein (GFAP), RPE-specific protein 65kDa (RPE65), pan-cytokeratin (pan-CK) and cytokeratin 18 (CK18). Hematoxylin staining showed no nuclei in the center of the degenerative lesion, thus suggesting the possibility of the occurrence of apoptosis. In the degenerative lesion specimen, GFAP staining was observed in the center, RPE65 staining was observed in the slightly peripheral region, and pan-CK staining was observed in all areas. However, no obvious staining with CK18 was observed. In a monkey retina used as the control specimen of a normal healthy retina, no RPE65 or pan-CK staining was observed in the neural retina. Our findings suggest that migration, proliferation, and differentiation of RPE cells might be involved in the repair of retinal lattice degeneration." (Lines 18-35)
(4)In the introduction, there are no details about RPE65 and CK and about the motivation of why the authors used them and which cell-types are labeled by them in the healthy retina? Please give some relevant background to help the readers to understand these.
Response to Comment 4: We greatly appreciate the Reviewer’s comment. In accordance with the Reviewer’s suggestion, please note that we have now added the following text regarding RPE65 and CK in the Introduction section:
"Retinal pigment epithelium-specific protein 65kDa (RPE65) is a specific marker of RPE cells and is reportedly involved in retinoid metabolism, one of the functions of retinal pigment epithelial cells [8]. Cytokeratin (CK) is one of the intermediate filaments that form the cytoskeleton of epithelial cells, and it is specifically expressed in the retinal pigment epithelium [9]." (Lines 60-64)
(5)In the results, the first part (2.1) mostly belongs to the materials and methods section.
Response to Comment 5: We greatly appreciate the Reviewer’s helpful comment. In accordance with the Reviewer’s suggestion, please note that we have now revised the Results section to read as follows:
"2.1. Collection of the retinal lattice degeneration specimens
In both cases, retinal lattice degeneration specimens at the approximate width and length of 1 mm and 3 mm, respectively, were collected (Fig 1a,b), and the papilla diameter was approximately 1.5 mm in the two cases. No intraoperative complications were observed during the collection of the lesions. After resection of the peripheral retinal lattice degeneration, pneumatic retinopexy, endophotocoagulation, and a gas tamponade with 20% sulfur hexafluoride was performed. In both cases, the retina was attached after surgery, and the postoperative courses were favorable." (Lines 71-78)
(6)In Results 2.2 the central cells did not have nuclei. Were those dead cells?
Response to Comment 6: We greatly appreciate the Reviewer’s comment. In accordance with the Reviewer’s question, please note that we have now added the following text in the Results section:
"2.2. Hematoxylin staining of retinal lattice degeneration
Although no nuclei were observed in the center of each retinal lattice degeneration specimen, diffuse nuclei were detected in the peripheral regions (Figure 2, blue arrowheads). However, no nuclei were observed in the central cells (Figure 2, black arrowheads)." (Lines 80-84)
Moreover, please note that the findings were already speculated in the Discussion section as follows:
"Hematoxylin staining revealed no nuclei in the center of the retinal lattice degeneration, while scattered nuclei were observed in the periphery. The absence of nuclei in the central lesion, where GFAP was intensely stained as described above, might have indicated the occurrence of apoptosis of the proliferated glial cells." (Lines 154-157)
(7)At L70. the authors state that the central cells are more intensely stained. It is not clear from the beginning that these stained specimens are slices or whole-mounted specimens. If the latter is true it is possible that the central area is thicker, thus having more dense staining.
Response to Comment 7: We greatly appreciate the Reviewer’s comment. Please note that those stained specimens were thick slices, so it is possible that the central cells were more intensely stained.
(8)There is no quantification of the staining densities in the paper. What is visible for the naked eye will also serve as a good base for quantification. Please quantify these (densities, cell numbers).
Response to Comment 8: We greatly appreciate the Reviewer’s instructive comment. In accordance with the Reviewer’s suggestion, please note that we have now added the following text in the Results section:
"The number of nuclei that stained positive for RPE65 in an area of 50μ2 was approximately 5 to 6. However, the number of nuclei that stained positive for pan-CK in that same 50μ2 area was approximately 20 to 30." (Lines 97-100)
9)What is the negative control of Fig3?
Response to Comment 9: We greatly appreciate the Reviewer’s comment. In accordance with the Reviewer’s question, please note that we have now added the following text in the Methods section:
"For comparison, we also created a negative control without these antibodies." (Lines 289 and 290)
10)What is visible in Fig5? There are no details about it in the text nor the figure legend. How it is related to lattice degeneration? The authors miss to correctly link any relation between the two.
Response to Comment 10: We greatly appreciate the Reviewer’s comment. In accordance with the Reviewer’s comment, please note that we have now added the following text in the Methods section:
"Since normal healthy human eye tissue was not available, we examined the localization of RPE65 and CK in the neural retina tissue obtained from a cynomolgus monkey eye." (Lines 293-295)
11)In the discussion, the authors assume that GFAP+ cells are activated Müller cells but they look more like astrocytes. Could they be part of a detached ILM? There is no real basis to draw such conclusions from these results on Müller cells, although RPE migration provides a feasible explanation for the RPE65/CK stainings, that I could agree with but it needs to be further investigated.
Response to Comment 11: We greatly appreciate the Reviewer’s instructive comment. As the Reviewer stated, these GFAP-positive cells could be astrocytes as well as Mueller cells. Please note that we have now revised the text to read as follows:
"GFAP is a marker of glial cells such as reactive Muller cells and astrocytes in the adult human retina [16]. Therefore, glial cells like Muller cells or astrocytes surrounding the retinal lattice degeneration might migrate, proliferate, and replace the damaged tissues."
(Lines 150-153)
12)Please define papilla diameter, since it is not widely used by non-clinicians.
Response to Comment 12: We greatly appreciate the Reviewer’s instructive comment. In accordance with the Reviewer’s suggestion, please note that we have now added the papilla diameter in the Results section of Fig.1.
"... and the papilla diameter was approximately 1.5 mm in the two cases." (Lines 73 and 74)
13)The authors should add product codes for the antibodies consequently. I could not find the codes for GFAP antibodies. Were these the same from two different companies? Were these rabbit antibodies? Have you tested the GFAP antibodies on retinal whole mounts as well to show astrocyte staining?
Response to Comment 13: We greatly appreciate the Reviewer’s instructive comment. In accordance with the Reviewer’s suggestion, please note that we have now added the information in the Methods section, as follows.
"GFAP staining was performed with a 1:100 dilution of the primary rabbit antibody (bs-0199R-A647; Thermo Fisher Scientific, Inc., Fremont, CA, USA or BioLegend®, Inc. San Diego, CA, USA), RPE65 staining was performed with a 1:200 dilution of the primary anti-RPE 65 rabbit antibody (401.8B11.3D9; Abcam, Cambridge, UK), pan-CK staining was performed with a 1:100 dilution of the primary anti-pan-CK rabbit antibody (C-11) (ab7753; Abcam), and CK18 staining was performed with a 1:100 dilution of the primary anti-CK18 rabbit antibody (C-04) (Abcam)." (Lines 272-278)
14)The procedure of slicing and embedding of the samples is not evidently clear. For example the mounting with Vectashield and slicing afterward. Which samples were whole mounts and which were slices? Please indicate it on the figures or figure legends.
Response to Comment 14: We greatly appreciate the Reviewer’s helpful comment, and we apologize for the confusion. In accordance with the Reviewer’s suggestion, please note that we have now added the following revisions in the Methods section and in the legends of Figures 2-4:
"Thin sections were then prepared for hematoxylin staining by slicing the specimen parallel to the retinal surface, and whole mounts were prepared for GFAP, RPE65, pan-CK, and CK18 immunostaining." (Lines 262-264)
"Figure 2. Hematoxylin staining of the obtained retinal lattice degeneration specimen via slicing parallel to the retinal surface. No nucleus was observed in the center of the retinal lattice degeneration (black arrowheads), yet diffuse nuclei were detected in the peripheral area (blue arrowheads).
Figure 3. GFAP, RPE65, pan-CK, and CK18 immunostaining of the retinal lattice degeneration whole-mount specimen in Case 1. The tissue was stained positive for GFAP, and the central area was slightly more intensely stained (black arrowheads) (a). For RPE65, the entire tissue was stained positive, while the margins of the central area without nuclei were more intensely stained (black arrowheads) (b). For pan-CK, the tissue was almost consistently stained positive (c), whereas it was little stained with CK18 (d). Negative control (e).
Figure 4. RPE65 and pan-CK immunostaining of the retinal lattice degeneration specimen in Case 2. The whole-mount tissue was stained positive for RPE65, whereas the central area with a few nuclei was less intensely stained (black arrowheads) (a). For pan-CK, the tissue was almost consistently stained positive (b). Negative control (c)."
(Lines 493-509)
15)Please recheck the methods section to maintain possible reproductibility.
Response to Comment 15: We greatly appreciate the Reviewer’s helpful comment. Please note that we have now rechecked the entire Methods section in order to maintain possible reproducibility.
16)Other comments
L15. The specimens
L113. RPE cells are involved…
L130. GFAP is a marker of
L139. immunostaining
L148. These pieces of evidence
L149. which were
L178. of the internal
Response to Comment 16: We greatly appreciate the Reviewer’s helpful comment. Please note that we have now corrected all of the typographical errors.
Round 2
Reviewer 1 Report
Most of the experiments asked in the review were not performed due to the limited availability of tissue but the authors tried to include the limitations in their results and discussion.
Author Response
Response to Comment: We greatly appreciate the Reviewer’s comment. In accordance with the Reviewer’s suggestion, please note that we have now added the following sentence in the Discussion section:
"It should be noted that in the present study, only a limited number of specimens were available for testing, so additional experiments are needed in the future." (Lines 246 and 247)

Reviewer 2 Report
The article advanced greatly since the last version. Now it gives a much clearer picture of how the experiments were performed and of the background of the experiments. With these relevant changes, it could be accepted, however, there are some remaining issues that need to be corrected before the final version.
Just two comments: I still think the abstract is far from perfect (it has been supplemented in a way and not reorganized).
Most points in the study still remain speculation, but showing this new data on lattice degeneration in the human eye is essential and interesting.
Remaining issues
- I cannot see any black arrowhead in Fig1.
- L95. µ2 is not SI unit. Please use SI (µm2).
After correcting these I see no obstacles in the way to accept this article.
Author Response
Reviewer 2’ comments
The article advanced greatly since the last version. Now it gives a much clearer picture of how the experiments were performed and of the background of the experiments. With these relevant changes, it could be accepted, however, there are some remaining issues that need to be corrected before the final version.
Just two comments: I still think the abstract is far from perfect (it has been supplemented in a way and not reorganized).
Response to Comment: We greatly appreciate the Reviewer’s helpful comment, and we greatly apologize for the poor readability and formatting (organization) of the Abstract. Please note that we have now rewritten the Abstract for improved readability and clarity, and to organize the sections in a more correct format that follows the MDPI guidelines (single paragraph, unstructured) as follows:
"Abstract
Lattice degeneration involves thinning of the retina that occurs over time. Here we performed an immunohistological study of tissue sections of human peripheral retinal lattice degeneration to investigate if retinal pigment epithelium (RPE) cells are involved in the pathogenesis of this condition. In two cases of retinal detachment with a large tear that underwent vitreous surgery, retinal-lattice-degeneration tissue specimens were collected during surgery. In the obtained specimens, both whole-mount and horizontal-section slices were prepared, and immunostaining was then performed with hematoxylin and antibodies against glial fibrillary acidic protein (GFAP), RPE-specific protein 65kDa (RPE65), pan-cytokeratin (pan-CK), and CK18. Hematoxylin staining showed no nuclei in the center of the degenerative lesion, thus suggesting the possibility of the occurrence of apoptosis. In the degenerative-lesion specimens, GFAP staining was observed in the center, RPE65 staining was observed in the slightly peripheral region, and pan-CK staining was observed in all areas. However, no obvious CK18 staining was observed. In a monkey retina used as the control specimen of a normal healthy retina, no RPE65 or pan-CK staining was observed in the neural retina. Our findings suggest that migration, proliferation, and differentiation of RPE cells might be involved in the repair of retinal lattice degeneration." (Lines 19-36)
Most points in the study still remain speculation, but showing this new data on lattice degeneration in the human eye is essential and interesting.
Remaining issues
1).I cannot see any black arrowhead in Fig1.
Response to Comment 1: We greatly appreciate the Reviewer’s helpful comment, and we greatly apologize for the oversight on our part. Please note that we have now added the needed black arrowheads in Figure 1a and Figure 1b.
2.L95. µ2 is not SI unit. Please use SI (µm2).
Response to Comment 2: We greatly appreciate the Reviewer’s helpful comment, and we apologize for the typographical error. Please note that we have now replaced "µ2" with "µm2" in the two associated sentences, as follows:
"The number of nuclei that stained positive for RPE65 in an area of 50μm2 was approximately 5 to 6. However, the number of nuclei that stained positive for pan-CK in that same 50μm2 area was approximately 20 to 30." (Lines 98-101)
After correcting these I see no obstacles in the way to accept this article.
